# Simulation of Saltwater Intrusion in the Minho River Estuary under Sea Level Rise Scenarios

Guilherme Menten [1], Willian Melo [1], José Pinho [1,*], Isabel Iglesias [2] and José Antunes do Carmo [3]

1   Centre of Territory, Environment and Construction (CTAC), Department of Civil Engineering, University of Minho, 4704-553 Braga, Portugal; gsmenten@gmail.com (G.M.); willianwm94@gmail.com (W.M.)
2   Interdisciplinary Centre of Marine and Environmental Research (CIIMAR/CIMAR), University of Porto, 4450-208 Matosinhos, Portugal; iiglesias@ciimar.up.pt
3   Faculty of Science and Technology, University of Coimbra, 3030-788 Coimbra, Portugal; jsacarmo@dec.uc.pt
*   Correspondence: jpinho@civil.uminho.pt; Tel.:+351-253-604-720

**Abstract:** Estuaries are areas that are vulnerable to the impacts of climate change. Understanding how these impacts affect these complex environments and their uses is essential. This paper presents a work based on an analytical solution and 2DH and 3D versions of the Delft3D numerical model to simulate the Minho River estuary and its saline wedge length under climate change projections. Temperature observations at several locations in the estuary region were selected to determine which model better simulated the temperature patterns. Specific simulations were performed for the observation periods. Sixteen numerical model scenarios were proposed, considering a varying tide, different river flows, and several SLR projections based on the RCP4.5 and RCP8.5 for 2050 and 2100. The analytical solution was also calibrated using the numerical model solutions. The results show that although there is no relevant stratification, there was a difference in both models in which in the worst climate change scenario, the length of the saline intrusion increased up to 28 km in the 2DH model and 30 km in the 3D model. It was concluded that the 3D model results were more precise, but both configurations can provide insights into how the saline intrusion will be affected. Additionally, the excellent agreement between the analytical solution and the results of the numerical models allowed us to consider the analytical solution a helpful tool for practical applications. It was demonstrated that freshwater discharges and bed slopes are the most critical drivers for the saline intrusion length in the Minho River estuary as they have more impact than the increase in sea level. Therefore, flow regulation can be an excellent way to control saline intrusion in the future.

**Keywords:** climate change; Delft3D; saline intrusion; Minho River estuary

## 1. Introduction

In recent decades, the impacts of climate change have been observed worldwide, such as changes in the mean sea level due to ice melting and the thermal expansion of the water masses associated with global warming and precipitation patterns due to variations in the water cycle and atmospheric circulation [1]. These effects are spatially and temporally heterogeneous, so it is necessary to apply local studies to understand the specific repercussions for different regions and periods [2].

Estuaries and coastal areas are some of the most productive ecosystems in the world, providing numerous environmental and economic benefits. These regions are also ideal for the installation of ports and shipyards. However, they are considered among the areas most vulnerable to the effects of climate change [3]. Estuaries are complex systems in which physical, geological, chemical, and biological components are interconnected. Therefore, simple measurements such as salinity and temperature can be critical indicators for assessing ecosystem changes [4].

Mixing processes between freshwater flows and oceanic salinity can be crucial in defining estuaries' structural and functional characteristics. The higher-density saline ocean

waters can penetrate the bottom layer of the estuaries and produce strong salinity/density gradients. This "salt-wedge" effect induces currents and can influence mixing conditions, sediment resuspension, and circulation patterns [5].

A higher sea level can cause saline water to migrate further upstream, potentially influencing the availability of water suitable for agricultural and industrial uses and retaining the sediments in the lower river courses. This can occur periodically due to high water levels from coastal storms (storm surge) or permanently due to climate change conditions. The permanent increase in sea level can also affect an estuary's response to extreme floods and waves. It can produce changes in the tidal range, increasing the tidal velocities and altering vertical mixing and erosion/accretion patterns [6–8].

As shown by several studies, changes in storm waves can potentially cause coastal impacts, which seem to be riskier than those related to sea level rise (SLR). On the Portuguese Atlantic coast, waves often reach significant heights of 2 m to 3 m. Significant heights of 9 m or more are achieved during storms, and such conditions can persist for up to 5 days [9]. Examples include the storms Rafael, Hercules, Stephanie, and Joaquin, which occurred in October 2012, January 2014, February 2014, and September 2015, respectively [10]. Changes in the frequency and intensity of extreme weather events will likely have notable effects on the Portuguese northwestern coast. This is reinforced because on this coast, many estuaries with a considerable number of exposed areas that are vulnerable to coastal storm effects can be found, notably, the Minho, Lima, Cávado, Douro, Aveiro, and Mondego estuaries [11]. Many Portuguese estuaries and the associated rivers are surrounded by low-lying agricultural and fish-farming areas vulnerable to coastal storms and changes in saltwater intrusion, so it is of the utmost importance to perform local studies to understand the future patterns of saltwater intrusion and to provide accurate information to policymakers to reduce the vulnerability of the estuarine regions.

In addition to waves, during storms, estuarine systems can also have short-lived episodic freshwater flows that practically fill the estuary with freshwater, blocking the entrance of saltwater into the estuarine system. River flow changes can alter tidal penetration, hydrodynamics, mixing, sedimentation patterns, nutrient delivery, primary production, the residence time of the water masses, and environmental conditions [12]. Persistent changes in freshwater flows and sea levels can lead to changes in estuarine classification, moving from a highly stratified salt wedge to a partially mixed estuary and/or to a vertically homogeneous estuary [13]. These changes in salinity gradients may affect water quality and the marine and freshwater organisms most sensitive to salinity thresholds.

Therefore, the sustainable management of coastal and estuarine areas needs precise technical and scientific information to be available to policymakers to promote the resilience of these areas and mitigate risks [7,14]. Assessing such impacts of climate change in estuarine dynamics must be fast and accurate enough to allow researchers to investigate different scenarios and update them with the newest predictions and societal changes. Therefore, numerical models have been extensively used in estuarine and coastal studies [15–17]. Among all processes that the models can simulate, the intrusion of seawater into estuaries and upstream rivers is one, incorporating complex bathymetries and detailed descriptions of turbulent exchange processes that allow for a proper representation of this phenomena, as shown in Chen et al. [14], Bigalbal et al. [18], and Ospino et al. [19], among others.

However, in addition to numerical models, analytical solutions can also be used to estimate saltwater intrusion, providing essential insights with lower computational needs than numerical models. The first analytical approach to the salt-wedge length was derived by Schijf and Schönfled [17]. These authors considered a one-dimensional, two-layer shallow water flow in prismatic channels with constant widths and assumed a constant seawater level and river inflow. Geyer and Ralston [20] and Krvavica et al. [21] provided recent developments in these solutions, among others. Particularly, Krvavica and Ružic [22] improved the theoretical solution by extending the classical solution for sloped channels, although with constant widths. However, it must be stressed that in neglecting several processes that contribute to seawater intrusion, the classical analytical solution is a strong simplification of reality. Considering

a prismatic channel and a tidal average, this solution neglects important seawater intrusion mechanisms such as (i) dispersion by shear effects and horizontal residual circulations; (ii) dispersion related to lateral exchange flows induced by secondary currents and channel bends; (iii) longitudinal dispersion induced by lateral baroclinic exchange flows, which is also favored by flood and ebb cycles; (iv) differential salinity advection due to baroclinic lateral currents (density-driven); and (v) dispersion related to the chaotic nature of the water particles' trajectories. Due to these limitations, the results of analytical solutions must be carefully analyzed for each case study to ensure its suitability.

Saltwater intrusion is a specific estuarine problem that will be exacerbated by the predicted trends of climate change impacts. Therefore, fast and accurate modeling methodologies must be applied in each case, and this work contributes to assessing the performances of both complex and simple modeling approaches. The work presents two different simulation approaches to assessing seawater intrusion in the Minho River estuary via analytical and numerical modeling solutions. The numerical model was previously implemented by Melo et al. [2] to examine the impact of climate change on the Minho estuary from a broader perspective, with flood risk identification as the focal point. In this study, the solutions of the numerical model provided a more comprehensive understanding of seawater intrusion in the estuary. To achieve this objective, both 2DH and 3D implementations of the model were utilized and compared using temperature field measurements. The utility of the analytical solution was also demonstrated via a comparison with the numerical model's results.

## 2. Study Area and Methods

### 2.1. Study Area

The Minho River is an international river that forms a natural border between Portugal and Spain in its last 70 km, with its mouth located between Caminha in northern Portugal and La Guardia in the Galician region of Spain. The Minho River has a total length of 340 km, and its average discharge flow is 300 m$^3$/s, ranging from 100 m$^3$/s in the summer season to 500 m$^3$/s in the winter. While the river's flow depends on the precipitation in the region, it is significantly regulated by the dams along the river, with the Frieira Dam situated 80 km upstream from the river's mouth playing a critical role in this regulation [13].

The Minho River estuary is 40 km long shallow estuary that varies in width from 200 m to 2000 m, narrowing to just 300 m at its mouth [23] (Figure 1). The estuary presents a semidiurnal, high-mesotidal regime and can be considered a partially mixed system in which a salt wedge configuration can be formed when the oceanic water enters the system [24,25]. In addition, the Minho estuary is considered a less-impacted estuary and is often selected as a reference in ecotoxicological studies due to its hydromorphological characteristics, water quality, populations, pollution, and great diversity of habitats containing valuable species [26].

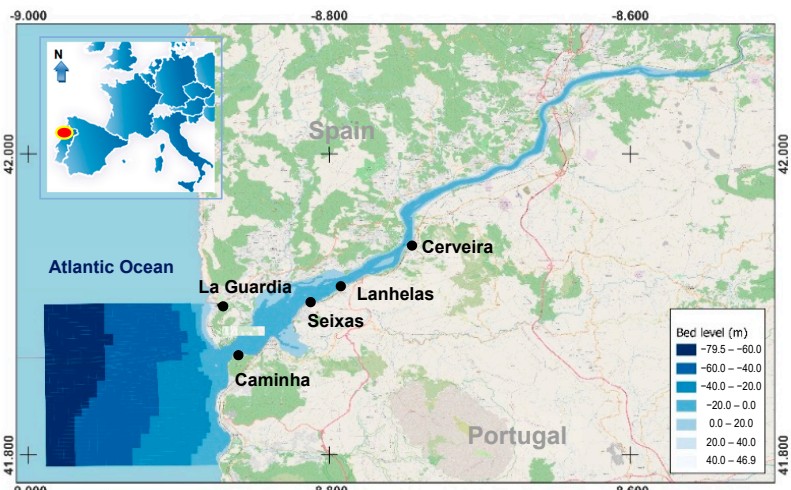

**Figure 1.** River Minho estuary location, model grid, and bathymetry.

### 2.2. Numerical Model

A detailed description of the model setup, calibration, and the approach used can be found in Melo et al. [2]. The numerical model selected was Delft3D software, which is widely used for simulating hydrodynamic processes in estuaries [26,27]. The software solves the Reynolds-averaged conservation equations of mass and momentum to accurately simulate estuarine hydrodynamics [2]. The momentum equations in both the $\widetilde{x}$- and $\widetilde{y}$-directions ($\sigma$−coordinates) are provided by (1) and (2), respectively, where *h* is the total water depth *(h = d + η)*, *d* is the water depth according to a reference level, and *η* is the variation in the water level:

$$\frac{\partial \widetilde{u}}{\partial t} + \widetilde{u}\frac{\partial \widetilde{u}}{\partial \widetilde{x}} + \widetilde{v}\frac{\partial \widetilde{u}}{\partial \widetilde{y}} + \frac{\widetilde{w}}{h}\frac{\partial \widetilde{u}}{\partial \sigma} = -\frac{1}{\rho_0}\left(\frac{\partial p}{\partial \widetilde{x}} + \frac{\partial \sigma}{\partial \widetilde{x}}\frac{p}{\sigma}\right) + f\widetilde{v} + F_x^v + \frac{1}{h^2}\frac{\partial}{\sigma}\left(v_v^t \frac{\partial \widetilde{u}}{\partial \sigma}\right) \quad (1)$$

$$\frac{\partial \widetilde{v}}{\partial t} + \widetilde{u}\frac{\partial \widetilde{v}}{\partial \widetilde{x}} + \widetilde{v}\frac{\partial \widetilde{v}}{\partial \widetilde{y}} + \frac{\widetilde{w}}{h}\frac{\partial \widetilde{v}}{\partial \sigma} = -\frac{1}{\rho_0}\left(\frac{\partial p}{\partial \widetilde{y}} + \frac{\partial \sigma}{\partial \widetilde{y}}\frac{p}{\sigma}\right) + f\widetilde{u} + F_y^v + \frac{1}{h^2}\frac{\partial}{\sigma}\left(v_v^t \frac{\partial \widetilde{v}}{\partial \sigma}\right) \quad (2)$$

In these equations, $F_x^v$ and $F_y^v$ represent the horizontal viscosity terms, which are dependent on the Reynolds stresses and satisfy the Boussinesq approach. In the previous equations, $\widetilde{u}$, $\widetilde{v}$ and $\widetilde{w}$ are the $\widetilde{x}$, $\widetilde{y}$ and $\widetilde{z}$ time-averaged velocity components in $\sigma$-coordinates, and $v_v^t$ is the vertical turbulent viscosity. The vertical momentum is reduced to the hydrostatic pressure distribution under the shallow water assumption. The continuity equation in $\sigma$-coordinates is provided by:

$$\frac{\partial \eta}{\partial t} + \frac{\partial h\widetilde{u}}{\partial \widetilde{x}} + \frac{\partial h\widetilde{v}}{\partial \widetilde{y}} + \frac{\partial \widetilde{w}}{\partial \sigma} = 0 \quad (3)$$

Geographic information system (GIS) techniques were used to delimit the model area, considering the region with an elevation of 4 m or less with respect to the Portuguese altimetric datum (mean sea level—MSL) [28,29]. Available bathymetric and topographic data were used to determine the model bathymetry [30].

The model used a calculation time step of 1 min, with a 60 min output interval. Manning's friction law was applied with a coefficient of 0.0155 m$^{-1/3}$·s, which was obtained from the model calibration. The estuary's latitude was assumed to be 43° N to consider the Coriolis effect.

The initial tidal harmonic constituents were derived from the TPXO 7.2 Global Inverse Tide Model in the Delft Dashboard and were later calibrated with field data. The calibration process was developed using OpenDA software that automatically obtains the best values of calibration parameters, considering the minimization of errors between simulated and observed hydrodynamic variables. The updated calibration values for the latest available field data used in the model and collected in 2021 are presented in Table 1.

**Table 1.** Main parameters used in the model, obtained from calibration.

| Parameter | Value | Parameter | Value |
|---|---|---|---|
| M2 amplitude (m) | 1.121 | N2 phase (degrees) | 57.341 |
| M2 phase (degrees) | 97.241 | K2 amplitude (m) | 0.102 |
| S2 amplitude (m) | 0.385 | K2 phase (degrees) | 102.124 |
| S2 phase (degrees) | 118.039 | Manning coefficient (m$^{-1/3}$s) | 0.0155 |
| N2 amplitude (m) | 0.226 | Horizontal viscosity/diffusivity(m$^2$/s) | 1 |

Aside from the open ocean boundary in which where the tidal constituents were imposed, an upstream fluvial open boundary condition was considered. The river water temperature was estimated by averaging data from Casais (Lat:+42.147 N, Lon:−8.203 W, station number 01H/03H from the SNIRH database, https://snirh.apambiente.pt/ accessed on 28 December 2021) and Miño-Salvaterra (Lat:+42.080 N, Lon:−8.496 W, station number

N015, from the Confederación Hidrográfica del Miño-Sil, https://www.chminosil.es/es/ accessed on 20 October 2021) monitoring stations during the dry season, resulting in a temperature of 19.5 °C [31]. The oceanic temperature was obtained from the Copernicus European Ocean-Sea Surface Temperature Multi-Sensor L3 Observations and L4 analysis, using the average temperature during the dry season, which was 16.6 °C.

The 3D version of the model was developed based on the 2DH model already developed by Melo et al. [4], considering ten vertical sigma layers for vertical discretization. The top and bottom layers were thinner compared to the intermediate layers to allow for better performances of the surface and bottom processes and therefore more accurate representations of the freshwater surface layer and the bottom salt-wedge entrance. The thickness percentages of the ten layers, considering the total water depth, were: 1st layer = 5%, 2nd layer = 5%, 3rd layer = 5%, 4th layer = 15%, 5th layer = 20%, 6th layer = 20%, 7th layer = 15%, 8th layer = 5%, 9th layer = 5%, and 10th layer = 5%.

### 2.3. Climate Change Scenarios

One of the main objectives of this study was to analyze the impacts of the SLR and variations in river flow on the extent of the saline intrusion, with particular attention provided to high-tide states during the spring tide.

To study saline intrusion, sixteen scenarios were considered by combining different river flow values and mean SLR values (see Table 2).

**Table 2.** Simulated scenarios.

| Scenario ID | River Flow (m$^3$/s) | SLR (m) |
|---|---|---|
| S1 | 10 | 0.00 |
| S2 | 10 | 0.21 |
| S3 | 10 | 0.53 |
| S4 | 10 | 0.24 |
| S5 | 10 | 0.77 |
| S6 | 20 | 0.77 |
| S7 | 30 | 0.77 |
| S8 | 40 | 0.77 |
| S9 | 50 | 0.77 |
| S10 | 100 | 0.00 |
| S11 | 100 * | 0.77 |
| S12 | 300 | 0.00 |
| S13 | 300 ** | 0.77 |
| S14 | 500 | 0.77 |
| S15 | 700 | 0.77 |
| S16 | 800 | 0.00 |

Note: * Dry season average flow; ** annual average flow.

Dams regulate the river flow in the Minho River hydrological basin, and the changes in the discharge patterns of the dams were considered as an alternative to hydrological modeling for estimating future flows. It was assumed that the dams' capacities for regulation and control would be maintained for future scenarios. To better understand the response of saline intrusion to and support future decisions regarding dam operation regulation, we mainly simulated low-river-flow conditions, including drought scenarios (Table 2, scenarios S1 to S9) and summer conditions (Table 2, scenarios S10 and S11). For comparison, additional scenarios with higher river flows were considered, including the annual averaged river flow (300 m$^3$/s, Table 2, scenarios S12 and S13) and higher river flows considering winter conditions (Table 2, scenario S14) and more extreme events (Table 2, scenarios S15 and S16). It is important to note that the region's projections point to drier conditions, especially during the summer, with an increase in the number and duration of dry events [32,33]. Although not directly correlated to the river's flow due to the dams, these conditions can increase pressure by decreasing the river's flow.

The impact of climate change on saltwater intrusion in the estuary was investigated through scenarios that considered SLR values for 2050 and 2100 using the RCP4.5 and RCP8.5 scenarios from the IPCC's Fifth Assessment Report, consistent with previous studies [2]. The study area is projected to experience relative SLRs of 21 cm (RCP4.5) and 24 cm (RCP8.5) by 2050 and 53 cm (RCP4.5) and 77 cm (RCP8.5) by 2100 [33].

All the considered scenarios incorporated the influence of tides to maintain proper mixing between saline and fresh waters in the estuary. A spin-up simulation period of two months was carried out with the current sea level conditions and an average river flow for both the 2DH and the 3D simulations to achieve a dynamic equilibrium state. The simulations began from a dry state with an initial water level of 0 m (MSL) and a uniform initial salinity of 18 ppt.

The outputs from the spin-up simulation were used as the initial conditions for the proposed scenarios, serving as the starting point for all subsequent simulations. These simulations also spanned two months.

For all scenarios considered, the maximum extent of saline intrusion was extracted for each simulation, allowing its comparison and analysis.

The results of the simulations were compared based on the salinity concentrations defined by Directive 2000/60/EC of the European Parliament and the Council of 23 October 2000, also known as the Water Framework Directive (WFD). Notably, the limits defined by the WFD are based on annual average concentrations, whereas this study focused on instantaneous concentrations. Therefore, those limits will only be used to compare the different scenarios. Therefore, following the WFD, the principal concentration limit discussed in this study will be 0.5 ppt, which is the concentration used to define saline intrusion. Salinity ranges are classified as follows: freshwater, 0 to 0.5 ppt; oligohaline, 0.5 to 5 ppt; mesohaline, 5 to 18 ppt; polyhaline, 18 to 30 ppt; euhaline, 30 to 40 ppt.

*2.4. Analytical Solution*

The classical theoretical approach for idealized flat estuaries was derived by Geyer and Ralston [20]. These authors considered a two-layer estuary separated by the pycnocline and assumed that the exchange between layers was slow when compared with advective processes within each layer and that the variations in the free surface height were negligible when compared with variations in the bottom depth. Also assuming that the width and depth of the estuary were constant, Geyer and Ralston obtained the following simplified solution (4):

$$\frac{\partial h_1}{\partial x} = \frac{F_1^2}{1 - F_1^2} C_f \left( 1 + \frac{h_1}{h_2} \right) \tag{4}$$

where $F_1 = q_1 / \sqrt{g h_1^3}$ is the internal Froude number for the upper layer $h_1$, g is the acceleration of gravity, $C_f$ is interfacial friction factor, and $h_2$ is the thickness of the lower layer (salt-wedge layer).

Following this approach, an analytical solution provided by Krvavica and Ružic [22] assumes that the channel geometry does not significantly affect the potential intrusion of the salt wedge but takes into account a non-uniform channel bed, $S_0$. It is defined as:

$$\frac{\partial h_p}{\partial x} = \frac{F_u^2}{1 - F_u^2} C_f \left( 1 + r \frac{h_p}{H_0 - S_0 x - h_p} \right) \tag{5}$$

where $h_p$ is the pycnocline depth, $F_u = q_r / \sqrt{g(1 - r) h_p^3}$ is the internal Froude number for the upper layer at each point of the grid, $H_0$ is the total water depth at the control section ($x_1 = 0$), $q_r$ is the river inflow rate per unit width, $C_f$ is the interfacial friction factor, O($10^{-3}$), $S_0$ is the channel bed slope, $r = \rho_1 / \rho_2$ is the density ratio, O(0.975), and $x$ is the dimensional salt-wedge length. It should be noted that approach (5) can be applied for different slopes along the river and therefore for any conditions with uneven bottoms.

Equation (5) is nonlinear for both unknown parameters ($h_p$ and $x$) and does not have an analytical solution, so it must be numerically integrated over the depth $h_p$. A predictor–corrector method may be used as follows:

$$h^*_{pi+1} = h_{pi} + \Delta x_i \times f(h_{Pi}, x_i) \tag{6}$$

$$h_{pi+1} = h_{pi} + \frac{\Delta x_i}{2}\left[f(h_{Pi}, x_i) + f\left(h^*_{pi+1}, x_{i+1}\right)\right] \tag{7}$$

with $f(h_p, x) = \left[\frac{F_u^2}{1-F_u^2}C_f\left(1 + r\frac{h_p}{H_0 - S_0 x - h_p}\right)\right]$ and $\Delta x_i = x_{i+1} - x_i$.

The boundary conditions, i.e., the lower and upper integral limits, are provided by $h_{P1} = h_{p,cr} = F_0^{2/3}H_0$, and $h_{Pn} = H_u$ (uniform flow height in the upstream section). The salt-wedge length $L$ is obtained by:

$$L = \sum_i^n \Delta x_i \tag{8}$$

Figure 2 shows the numerical solution (6)–(7) of the analytical approach (5) applied to three different channel bed slopes $S_0$ (0.0005, 0.00075, and 0.0010), considering $C_f = 0.0005$, $r = 0.975$, $H_0 = 8.0$ m, $h_{Pn} = 4.0$ m, $q_r = 0.5$ m$^2$/s, and $\Delta x = 5.0$ m.

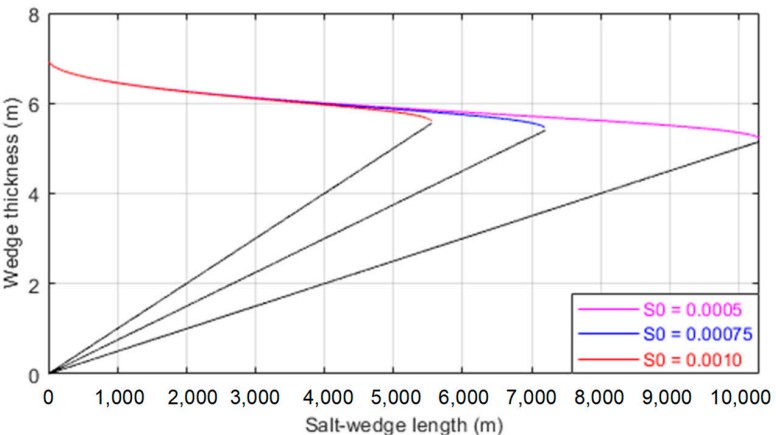

**Figure 2.** Salt-wedge intrusion $L$ as a function of internal Froude numbers $F_u$ for three different non-dimensional channel bed slopes $S_0$ (riverbed shown in black).

Saline intrusion lengths of 10,320 m, 7200 m, and 5555 m were obtained for bed slopes of 0.0005, 0.00075, and 0.0010, respectively.

As Figure 2 makes clear, the channel's bottom geometry significantly affects the potential intrusion of a salt wedge, demonstrating the inadequacy of the theoretical approach (4) derived for idealized flat estuaries and rivers.

### 2.5. Comparison of 2DH and 3D Models

Temperature measurements were used to compare the results of the numerical model with observations. It is important to note that this is not a verification or validation of the numerical model since it has already been validated [5]. Instead, the goal is to observe which model better simulates the temperature behavior of the estuary and compare the results from both the 2DH and 3D models.

Temperature observations were carried out using RBR submersible temperature and pressure loggers (TGR-2050 and TWR-2050) at four locations in the estuary: Caminha, Seixas, Lanhelas, and Cerveira (Figure 1). Specifically, temperature data were collected in Caminha and Seixas between 6 and 8 October 2021 and in Lanhelas and Cerveira between 11 and 15 October 2021. Additional data were collected in Cerveira from 5 November to 9 December 2021 and in Lanhelas from 19 November to 9 December 2021.

Specific model simulations were performed for the observational periods. For these simulations, river flow conditions were extracted from data measured at Foz do Mouro (SNIRH, https://snirh.apambiente.pt/ accessed on 28 December 2021), and tidal conditions were included at the oceanic boundary through the application of local main tidal constituents. At the fluvial boundary, the temperatures measured by monitoring stations for the same periods were assumed [28–30]. Ocean boundary temperatures were defined based on Copernicus data. However, since the Copernicus temperature data are measured at the surface layer, we applied an offset to each temperature time series to account for the expected lower temperature of the seawater that enters the estuary. To estimate the appropriate temperature offset, we used data measured at Caminha for reference and applied the same offset value to the other locations of comparison.

## 3. Results

This paper discusses two main results: first, the comparison of the 2DH and 3D models for simulating temperature and saltwater intrusion in the Minho River estuary, in addition to an analytical solution for the salt-wedge length; second, the effects of climate change scenarios on the saline intrusion of the estuary.

### 3.1. Temperature Simulations Results

The 2DH and 3D models were ablet o simulate the temperature variation behavior during the studied period, as demonstrated in Figure 3.

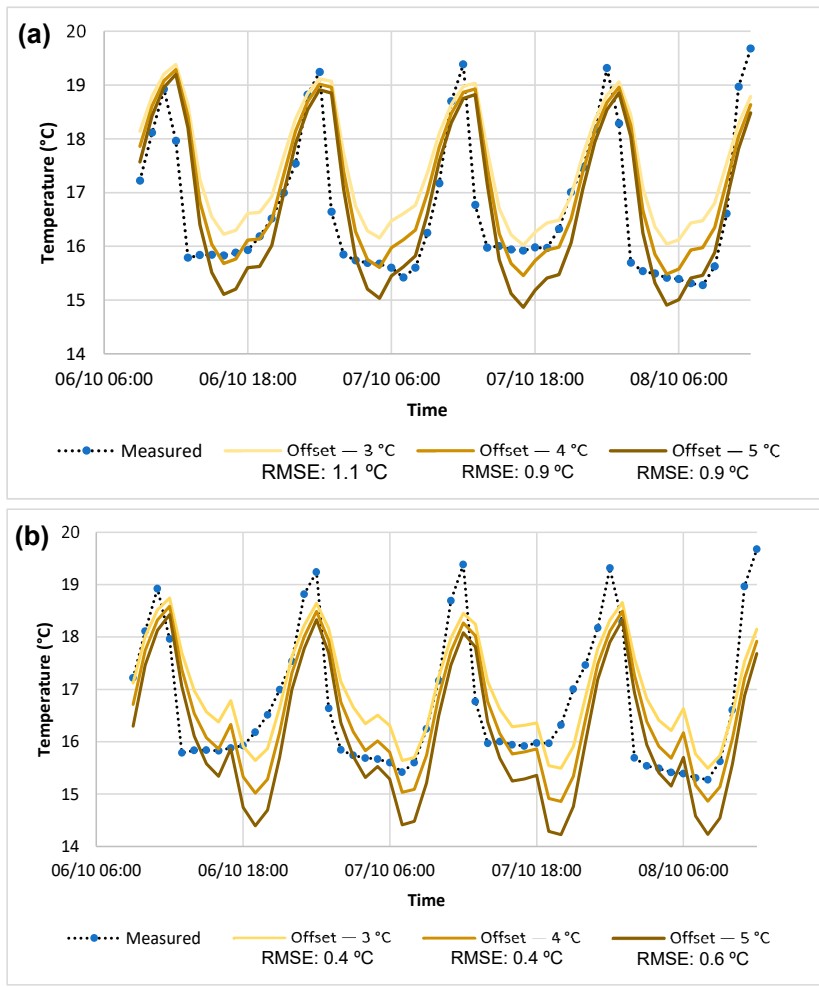

**Figure 3.** Simulation results with different ocean boundary temperature adjustments compared with measured field data. (**a**) 2DH model; (**b**) 3D model.

The simulation results show that, while the 2DH and 3D models have slight differences in behavior, they reproduce the main temperature pattern induced by the tide along the estuary. Based on the root mean squared error (RMSE) results shown in Figure 3, an offset of −4 °C was selected for the other locations. The 2DH model presented a RMSE of 0.9 °C, while the 3D model had a lower RMSE of 0.4 °C, indicating that the 3D model results are closer to the observed values. However, the difference between them is not significant.

Figure 4 displays the results obtained for other locations, demonstrating that both models produced similar outcomes, with RMSE values around 1 °C.

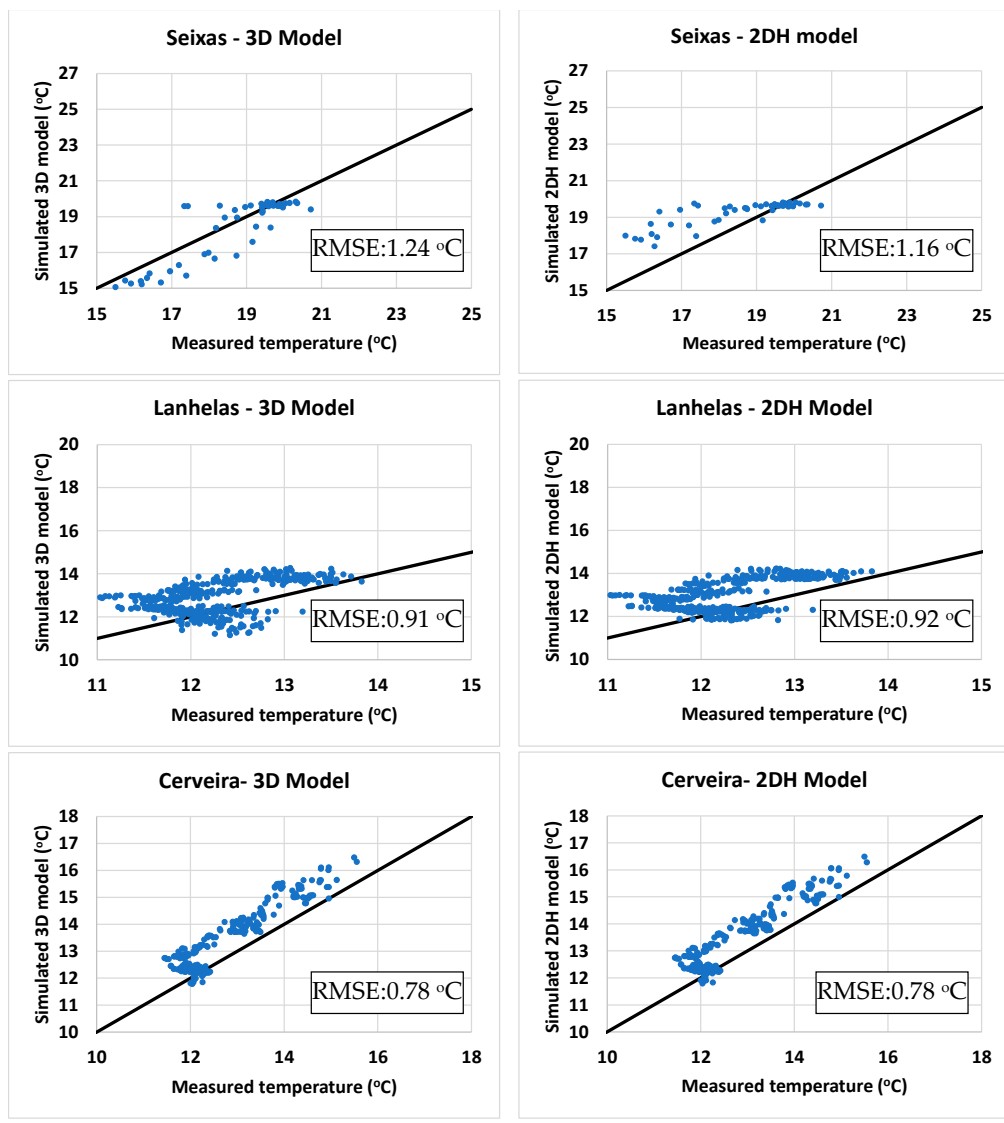

**Figure 4.** Comparison between measured water temperature and results from 2DH and 3D models in Seixas, Lanhelas, and Cerveira for 2DH and 3D models.

## 3.2. Estuarine Stratification Conditions Analysis

Given the limited depth of the estuary and the previous results obtained, the 2DH model can be considered enough and appropriate for the objectives of the saline intrusion extension analysis. This work also achieved similar findings as Melo et al. [2], who employed a 3D model but did not observe any significant vertical stratification. Results from a simulation using a 3D model but with six vertical sigma layers, as presented in Figure 5, further support adopting a 2DH.

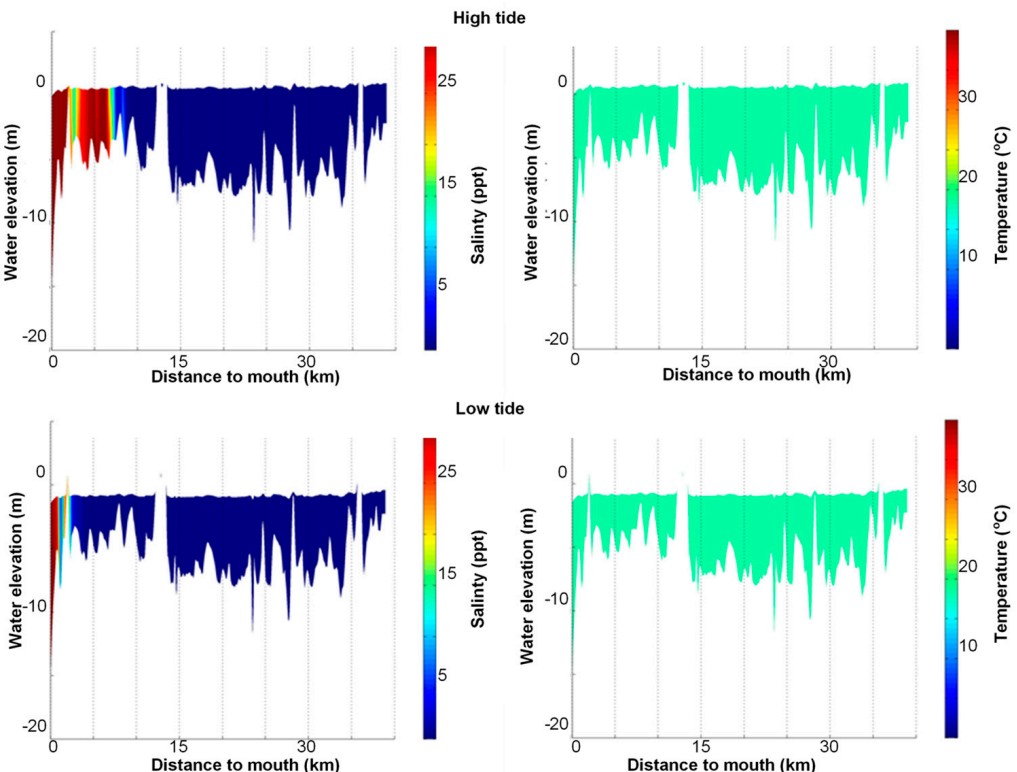

**Figure 5.** Longitudinal profiles in a scenario with a flow of 325.82 m$^3$/s and without SLR (adapted from Melo et al. [2]).

In this study, the 3D model simulations also reveal no stratification along the estuary for average river flow conditions of around 300 m$^3$/s. Considering the last month of each simulation, the results for three locations were processed to calculate the averaged salinity concentration differences between the upper and lower layers of the model. No stratification was represented by the numerical model for the three selected locations. The maximum salinity difference observed for all sites was 0.006 ppt. This maximum difference was obtained at the river's mouth, and the difference diminished in the upstream direction (Figure 6).

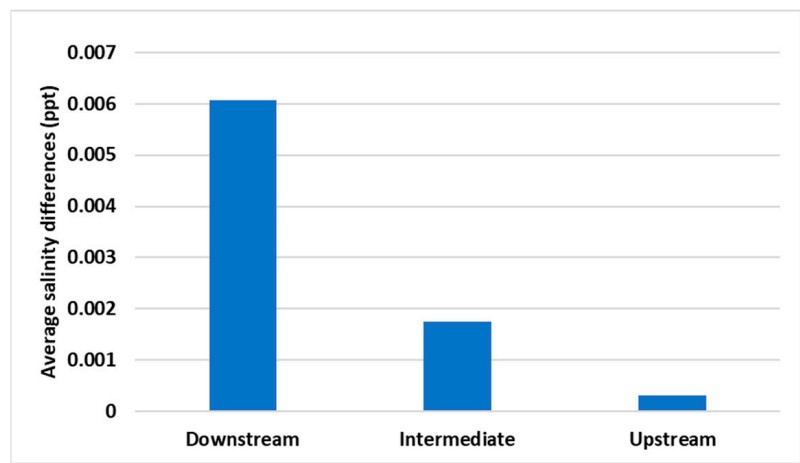

**Figure 6.** Average differences between the upper and the lower layers of the 3D model at three different locations for a one-month simulation considering tide action and a river flow of 300 m$^3$/s.

### 3.3. Analysis of Saline Intrusion

The extent of saline intrusion in each considered scenario and numerical model (2DH and 3D) is represented in Figures 7 and 8.

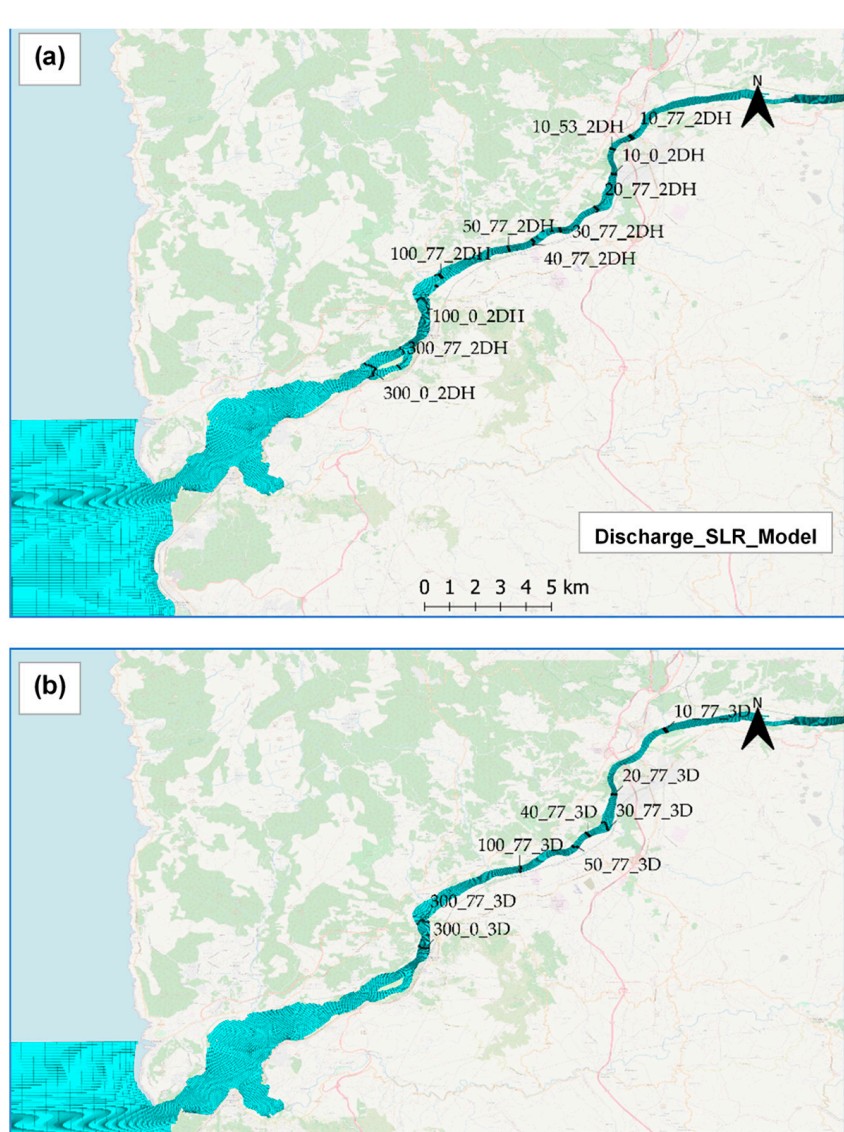

**Figure 7.** Saline intrusion extent, considering a 0.5 ppt concentration limit. Labeled as the river flow discharge (m$^3$/s), SLR (cm), and model applied: (**a**) 2DH model; (**b**) 3D model.

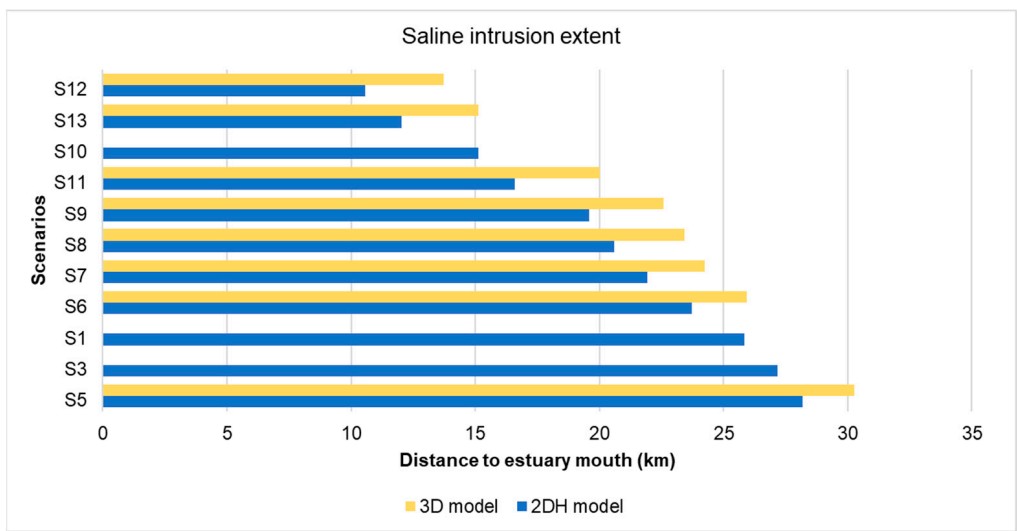

**Figure 8.** Extent of saline intrusion, with the estuary mouth taken as a reference for the simulated scenarios with both 2DH and 3D models.

Scenarios S2 and S3 simulated the SLR (RCP4.5) for the years 2050 and 2100, respectively, considering mean sea levels of 0.21 m and 0.53 m in severe drought conditions (Table 2). The extent of saline intrusion presented lengths of 27.7 km and 28.4 km, respectively, relative to the mouth. Scenarios S4 and S5 were associated with the estimated SLR for the RCP8.5 emissions scenario, predicting a mean SLR of 0.24 m in 2050 and 0.77 m in 2100. In these cases, the salt wedge reaches 27.8 km and 28.9 km from the mouth for the 2050 and 2100 scenarios (Figures 7a and 8).

For the remaining scenarios, the mean sea level was considered identical to the one estimated in the RCP8.5 emission scenario for 2100, while the river flow presented different values. Figure 8 summarizes the results of all simulated scenarios, highlighting the influence of the two factors analyzed: the river flow discharge and the mean SLR.

The 3D model's results show greater extents of saline intrusion than the 2DH model when comparing the same scenario (Figure 8), with the differences increasing in the scenarios with less saline intrusion, ranging from 7% in S5 to 30% in S12. The river flow discharges in both models were the main driver for increasing the extent of saline intrusion.

The findings reveal that the saline intrusion can extend 30 km from the river's mouth into the estuary in the 3D model or 28 km in the 2DH model. Both models demonstrate a nonlinear relationship between the length of the saline intrusion and the river's flow. Interestingly, the scenario with a flow rate of 10 $m^3/s$ and no SLR exhibits a larger saline intrusion than the scenario with a flow rate of 20 $m^3/s$ and the most pessimistic SLR scenario. This demonstrates that the river flow is the primary driver of saline intrusion for drier scenarios in the Minho River estuary. Therefore, regulating the river's flow seems to be a viable solution for counteracting the impact of SLR on saline intrusion.

Figure 9 provides a comprehensive overview of the results obtained from all simulated scenarios, focusing on the influence of the factors under investigation: (a) river flow and (b) mean SLR. The figure effectively highlights the impacts of these factors and their interplay on the observed outcomes, demonstrating the nonlinear relation between these factors and the extent of saline intrusion.

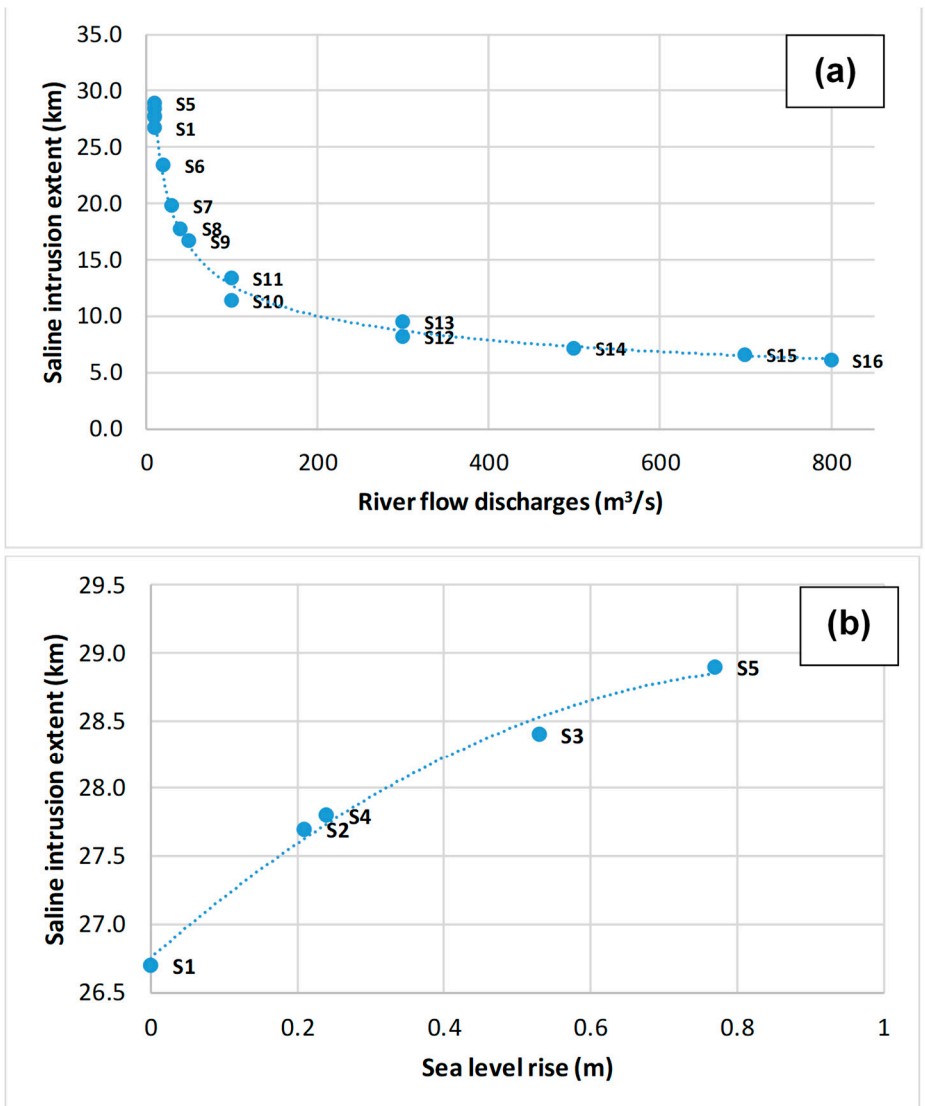

**Figure 9.** Results of saline intrusion in the Minho River estuary. (**a**) Extension of saline intrusion as a function of river flow; (**b**) extension of saline intrusion due to the increase in mean sea level, considering a minimum river flow (severe drought).

*3.4. Salt-Wedge Length in the River Minho Estuary Provided by the Analytical Solution*

Applying the analytical/numerical solution derived from the classical theoretical approach for idealized flat estuaries requires a proper definition and quantification of all the parameters involved. In this work, the average channel slope and estuarine width were defined assuming a linear slope between the depth at the control section and the upstream depth (Figure 10) and through the quotient between the surface area at mean sea level and the estuarine axis length, respectively (so = 0.0001 and width = 630 m).

Moreover, a density ratio $r = 0.975$ and a total water depth at the control section of $H_0 = 5.3$ m were adopted. Multiple computations were carried out to approximate the analytical/numerical solutions to the 2DH/3D results by varying the interfacial friction factor $C_f$ and adjusting the initial $h_p$ value at the control section to the actual river discharges for each scenario (Figure 11a). The values of the $C_f$ and initial $h_p$ parameters as functions of the river discharge for the Minho River estuary that produced the best results are presented in Figure 11b.

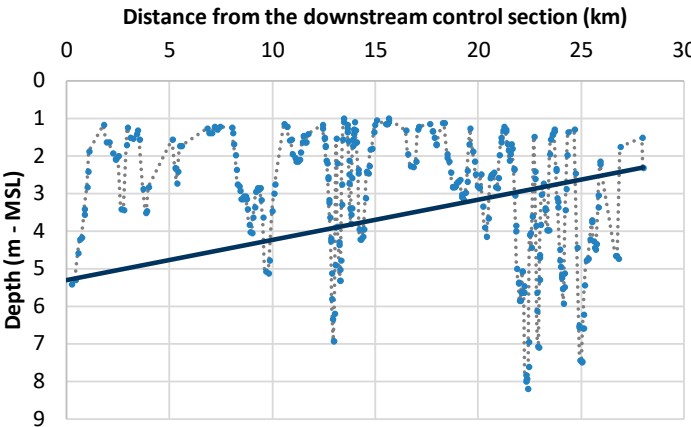

**Figure 10.** River Minho estuary bed slope quantification for the application of the analytical/numerical solution derived from the classical theoretical approach for idealized flat estuaries.

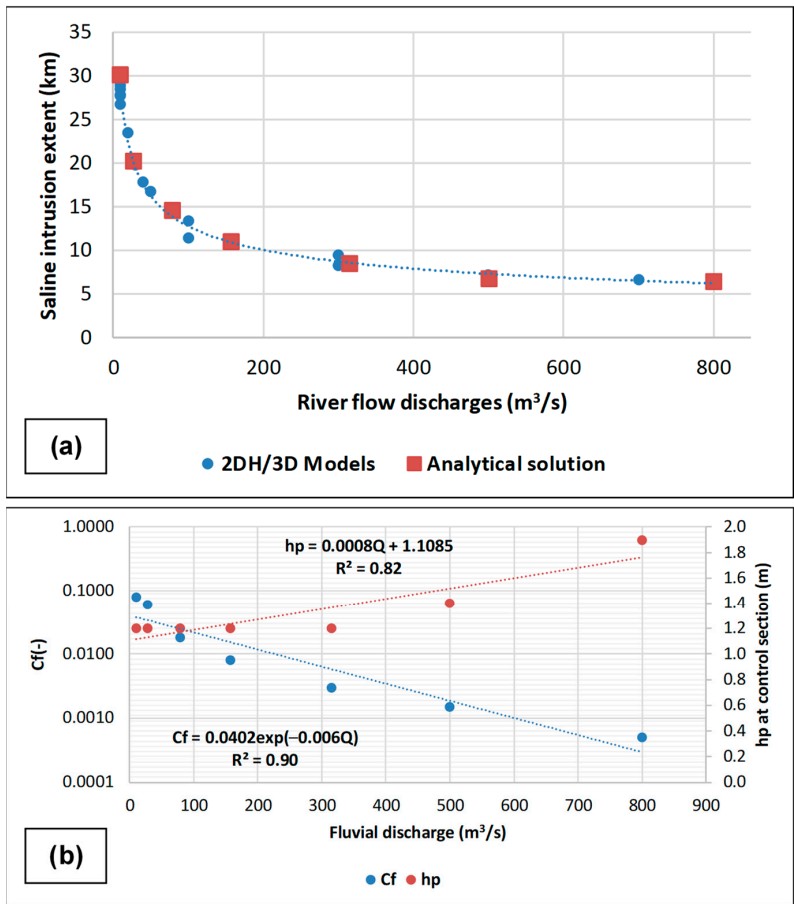

**Figure 11.** Results of saline intrusion in the Minho River estuary. (**a**) Extension of saline intrusion as a function of river flow obtained by both 2DH/3D models and calibrated analytical/numerical solution; (**b**) obtained $C_f$ and initial $h_p$ value functions of river discharge for the Minho River estuary.

Despite the simplifications considered in the analytical solution (2), the results shown in Figure 11 agree with the 2DH/3D numerical solutions. Therefore, the analytical solution (2) can be considered a useful, simple, and practical tool for the preliminary assessment of the saline wedge length in estuaries. It can also be used for short- to medium-term forecasts, presenting a higher computational efficiency than the complex 2DH/3D models to estimate saline intrusion in the Minho River estuary.

## 4. Discussion and Conclusions

This study demonstrated that the 2DH and 3D models could simulate the water temperature patterns influenced by tides and river flows, albeit with slight differences. The water temperature results revealed that the 3D model exhibited smaller errors; however, it was observed that the difference between the models decreased further upstream, where the length of the saline intrusion was determined. In this context, both models displayed poorer performances in the upstream area. This finding agrees with recent studies that also reported challenges in accurately modeling temperature patterns in estuaries (e.g., [34,35]).

Furthermore, the analytical solution emerged as a simpler and faster method for estimating saltwater intrusion along the estuary, providing an advantage over the numerical models. However, its application requires the correct definitions of equation parameters ($C_f$ and $h_p$) based on the results presented in Figure 11b. This successful implementation of the analytical solution relies on prior knowledge of fluvial discharge and the availability of similar parameter results from other estuaries [36].

The simulations conducted in this study confirmed the absence of stratification in the estuary of the Minho River, which is consistent with findings from previously published works (e.g., [37]). This suggests that the estuary's hydrodynamic characteristics are relatively homogeneous and do not exhibit significant vertical stratification.

The projections of sea level rise (SLR) demonstrated that considering the lowest river flow and the highest SLR, the saline intrusion could extend over 30 km. Importantly, it was demonstrated that river flow plays a crucial role in increasing the extent of saline intrusion, surpassing the significance of SLR. However, notable discrepancies were observed between the 2DH and 3D models, with variations of up to 30% in the length of the saline intrusion.

Furthermore, the good agreement observed between the analytical solution and the numerical model results underscores the practical utility of the analytical solution. This agreement supports its applicability for estuary management and decision-making processes, providing a reliable and efficient tool for estimating saline intrusion [36].

An important conclusion drawn from this study, with implications for future estuary modeling in the Minho River and other similar systems, is that the effects of climate change on precipitation patterns and river flows cannot be disregarded. These factors have a more significant influence on saline intrusion than sea level rise. This highlights the necessity of adopting a holistic approach to estuary management, including regulating dam operations, as river flow can either be the main contributor to increased saline intrusion or a means of controlling it. Recent studies have similarly emphasized the importance of considering the impacts of climate change on river flows and precipitation in estuarine management strategies (e.g., [34,37]).

Another significant aspect worth considering is the potential alteration to the riverbed slope resulting from river regularization works. As demonstrated by the analytical solution, even slight variations in the riverbed slope can substantially impact the length of the salt wedge. This finding highlights the need to carefully consider riverbed modifications and their potential consequences on saline intrusion dynamics [38].

This research contributes valuable insights for estuary management, facilitating informed decisions regarding mitigating saline intrusion and enhancing preparedness for future changes. By improving the resilience of the estuary, ecological considerations and the diverse range of estuarine uses can be effectively addressed (e.g., [2]).

**Author Contributions:** Conceptualization, G.M. and J.P.; Methodology, G.M., J.P., I.I. and J.A.d.C.; Software, J.P. and J.A.d.C.; Validation, G.M. and W.M.; Formal analysis, G.M., I.I. and J.A.d.C.; Investigation, G.M., W.M., I.I. and J.A.d.C.; Writing—original draft, G.M.; Writing—review & editing, W.M., J.P., I.I. and J.A.d.C. All authors have read and agreed to the published version of the manuscript.

**Funding:** The authors want to acknowledge the contract funds provided by the project EsCo-Ensembles (PTDC/ECI-EGC/30877/2017), co-financed by NORTE 2020, Portugal 2020, and the European Union through the ERDF and by FCT through national funds. This research was also supported by the Doctoral Grant SFRH/BD/151383/2021, financed by the Portuguese Foundation for Science and Technology (FCT), and with funds from the Ministry of Science, Technology and Higher Education, under the MIT Portugal Program. This research was also partially supported by the Strategic Funding UIDB/04423/2020 and UIDP/04423/2020 through national funds provided by the FCT—Foundation for Science and Technology and the European Regional Development Fund (ERDF). I. Iglesias also wants to acknowledge the FCT financing through the CEEC program (2022.07420.CEECIND).

**Data Availability Statement:** The data that support the findings of this study are available from the corresponding author upon reasonable request.

**Conflicts of Interest:** The authors declare no conflict of interest.

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
