# Peer review of "Simulation of Saltwater Intrusion in the Minho River Estuary under Sea Level Rise Scenarios"

_water, doi:10.3390/w15132313_

Round 1
Reviewer 1 Report
This paper studies the saltwater intrusion in the Minho River estuary using analytical and numerical modelling solutions. This work is interesting and provides us a more comprehensive understanding of seawater intrusion in the estuary. Before it can be published in the journal of Water, some revisions are needed, as shown below:
1) All the Figures in the manuscript are very ugly, and should be improved;
2) Part 2.2: the controlling equation of the numerical model should be provided in this manuscript;
3) Line 147: How does the author obtained the coefficient of 0.0155 m−1/3·s?
4) Table 1 gives the main parameters used in the model. How do the authors calibrate these parameters?
5) What is the boundary condition in the model?
6) For the analytical and numerical modelling solution, are the governing equations are same? What are the differences between the analytical and numerical solution?
7) The simulation results should be compared and validated with the measured results;
8) The authors should discuss the advantages and disadvantages of the numerical model.
Author Response
Reviewer 1
This paper studies the saltwater intrusion in the Minho River estuary using analytical and numerical modelling solutions. This work is interesting and provides us a more comprehensive understanding of seawater intrusion in the estuary. Before it can be published in the journal of Water, some revisions are needed, as shown below:
A: Thank you for your positive evaluation of the paper content.
1) All the Figures in the manuscript are very ugly, and should be improved;
A: The authors have improved the figures.
2) Part 2.2: the controlling equation of the numerical model should be provided in this manuscript;
A: The main equations solved by the Delft3D – Flow module were inserted in section 2.2
3) Line 147: How does the author obtained the coefficient of 0.0155 m−1/3·s?
4) Table 1 gives the main parameters used in the model. How do the authors calibrate these parameters?
A: A sentence was added to clarify the process of automatic calibration with OpenDA:
"The calibration process was developed using OpenDA software that allows to automatically obtain the best values of calibration parameters considering the minimization of errors between simulated and observed hydrodynamic variables."
5) What is the boundary condition in the model?
A: A sentence was added to clarify the considered open boundary conditions of the models:
"Besides the open ocean boundary condition where the tidal constituents were imposed an upstream fluvial open boundary condition was considered."
6) For the analytical and numerical modelling solution, are the governing equations are same? What are the differences between the analytical and numerical solution?
A: In the revised paper, both formulations are now presented according to suggestion 2).
7) The simulation results should be compared and validated with the measured results;
A: Since we have used measured water levels to calibrate the hydrodynamics automatically, we present the comparisons between measured and simulated water temperature.
8) The authors should discuss the advantages and disadvantages of the numerical model.
A: This topic was added to the discussion:
"Moreover, the analytical solution is a much simpler and faster method to estimate the saltwater intrusion along the estuary, which is an advantage. However, it only can be used after the involved equation parameters (Cf and hp) are correctly defined from results presented in Figure 11 b). This only can be successfully achieved if the fluvial discharge is known in advance and similar results of the parameters are available when applied to other estuaries."

Reviewer 2 Report
The authors have attempted to work on simulating the salt water for desalination using numerical models. However, the authors have not presented merits and demerits from the model in developing the simulations for the desalination of salt water from the sea.
The abstract is very confined and written precisely.
introduction seems presented with basic information but not focused on the problem definition and statement.
The recent literature review is presented in the work. The national and international status of work is also not demonstrated.
The introduction should be rewritten considering the latest literature, importance of work, and state of art of work.
The numerical model utilised in the study is elaborately described and good information is provided by the authors for the readers reference.
the results are well established and nicely presented.
but the graphical representations are with poor resolution and need to be focused to improve the quality of figures.
the interpretation from the results does not consistent or matched wiht that presented in the graphs. there are several inconsistencies that need to be addressed by the authors before the submission of revised paper.
For examples, figures 8 and 9, the interpretation is not agreement with that shown in figures and not consistent with the previous reports.
The discussion and conclusion section is very generally written. Authors should focus more in rewriting this section highlighting the main results and conclusions referred from this work.
English language should be refined moderately as at several instances correction of grammatically errors is required.
The citations in the reference list are old references and hence author should focus on this list in improving with the latest references.
English language should be refined moderately as at several instances correction of grammatically errors is required.
Author Response
Reviewer 2
The authors have attempted to work on simulating the salt water for desalination using numerical models. However, the authors have not presented merits and demerits from the model in developing the simulations for the desalination of salt water from the sea.
A: Thank you very much for the comments/questions.
The advantages and disadvantages of the models were clarified in the discussion.
The abstract is very confined and written precisely.
A: Thank you for your appreciation.
introduction seems presented with basic information but not focused on the problem definition and statement.
A: The studied problem was emphasized with the insertion of a new paragraph:
"Saltwater intrusion is a specific estuarine problem that will be exacerbated by the predicted trends of climate change impacts. Fast and accurate modelling methodologies must, therefore, be applied in each case and this work contributes to assess the performance of both complex and simple modeling approaches."
The recent literature review is presented in the work. The national and international status of work is also not demonstrated. The introduction should be rewritten considering the latest literature, importance of work, and state of art of work.
A: A sound revision of the introduction paper section was done.
The numerical model utilised in the study is elaborately described and good information is provided by the authors for the readers reference.
A: Thank you very much for the comment.
the results are well established and nicely presented.
A: Thank you very much for the comment.
but the graphical representations are with poor resolution and need to be focused to improve the quality of figures.
A: The figure's resolution was improved.
the interpretation from the results does not consistent or matched wiht that presented in the graphs. there are several inconsistencies that need to be addressed by the authors before the submission of revised paper. For examples, figures 8 and 9, the interpretation is not agreement with that shown in figures and not consistent with the previous reports.
A: Figures 8 and 9 were revised in order to be consistent.
The discussion and conclusion section is very generally written. Authors should focus more in rewriting this section highlighting the main results and conclusions referred from this work.
A: The main results and conclusions are highlighted in the revised discussions and conclusions paper section.
English language should be refined moderately as at several instances correction of grammatically errors is required.
A: A sound revision of the English language was done.
The citations in the reference list are old references and hence author should focus on this list in improving with the latest references.
A: 5 more recent references were inserted, mainly in the discussion paper section.

Reviewer 3 Report
Some sentences are too complicated and hard to understand.
Rework “Discussion and Conclusion” Chapter. Part of this chapter belong to “Study Area and Methods”.
I have posted some remarks in the PDF file.

Some sentences are too complicated and hard to understand.
Author Response
Reviewer 3
Some sentences are too complicated and hard to understand. Rework "Discussion and Conclusion" Chapter. Part of this chapter belong to "Study Area and Methods".
A: Thank you very much for your comments.
Discussion and conclusions paper section
I have posted some remarks in the PDF file.
A: Thank you for the revision. All the suggestions in the pdf file were implemented, including the improvement of figures.
